# Poly(Allylamine Hydrochloride) and ZnO Nanohybrid Coating for the Development of Hydrophobic, Antibacterial, and Biocompatible Textiles

**DOI:** 10.3390/nano14070570

**Published:** 2024-03-25

**Authors:** Nives Matijaković Mlinarić, Barbara Wawrzaszek, Klaudia Kowalska, Atiđa Selmani, Aleksander Učakar, Janja Vidmar, Monika Kušter, Nigel Van de Velde, Polonca Trebše, Andrijana Sever Škapin, Ivan Jerman, Anže Abram, Anamarija Zore, Eva Roblegg, Klemen Bohinc

**Affiliations:** 1Faculty of Health Sciences, University of Ljubljana, Zdravstvena Pot 5, 1000 Ljubljana, Slovenia; nives.matijakovic@zf.uni-lj.si (N.M.M.); polonca.trebse@zf.uni-lj.si (P.T.); anamarija.zore@zf.uni-lj.si (A.Z.); 2Institute of Chemical Sciences, Faculty of Chemistry, Maria Curie-Skłodowska University in Lublin, Pl. Maria Curie-Skłodowska 3, 20-031 Lublin, Poland; wawrzaszek.barbara@gmail.com (B.W.); klaudia.kowalska113@gmail.com (K.K.); 3Pharmaceutical Technology and Biopharmacy, Institute of Pharmaceutical Sciences, University of Graz, Universitätsplatz 1, 8010 Graz, Austria; atida.selmani@uni-graz.at (A.S.); eva.roblegg@uni-graz.at (E.R.); 4Jožef Stefan Institute, Jamova Cesta 39, 1000 Ljubljana, Slovenia; aleksander.ucakar@ijs.si (A.U.); janja.vidmar@ijs.si (J.V.); monika.kuster@ijs.si (M.K.); anze.abram@ijs.si (A.A.); 5National Institute of Chemistry, Hajdrihova Ulica 19, 1000 Ljubljana, Slovenia; nigel.van.de.velde@ki.si (N.V.d.V.); ivan.jerman@ki.si (I.J.); 6Slovenian National Building and Civil Engineering Institute, Dimčeva Ulica 12, 1000 Ljubljana, Slovenia; andrijana.skapin@zag.si; 7Faculty of Polymer Technology—FTPO, Ozare 19, 2380 Slovenj Gradec, Slovenia

**Keywords:** ZnO, nanoparticles, poly(allylamine hydrochloride), textiles, *Staphylococcus aureus*

## Abstract

In healthcare facilities, infections caused by *Staphylococcus aureus* (*S. aureus*) from textile materials are a cause for concern, and nanomaterials are one of the solutions; however, their impact on safety and biocompatibility with the human body must not be neglected. This study aimed to develop a novel multilayer coating with poly(allylamine hydrochloride) (PAH) and immobilized ZnO nanoparticles (ZnO NPs) to make efficient antibacterial and biocompatible cotton, polyester, and nylon textiles. For this purpose, the coated textiles were characterized with profilometry, contact angles, and electrokinetic analyzer measurements. The ZnO NPs on the textiles were analyzed by scanning electron microscopy and inductively coupled plasma mass spectrometry. The antibacterial tests were conducted with *S. aureus* and biocompatibility with immortalized human keratinocyte cells. The results demonstrated successful PAH/ZnO coating formation on the textiles, demonstrating weak hydrophobic properties. Furthermore, PAH multilayers caused complete ZnO NP immobilization on the coated textiles. All coated textiles showed strong growth inhibition (2–3-log reduction) in planktonic and adhered *S. aureus* cells. The bacterial viability was reduced by more than 99%. Cotton, due to its better ZnO NP adherence, demonstrated a slightly higher antibacterial performance than polyester and nylon. The coating procedure enables the binding of ZnO NPs in an amount (<30 µg cm^−2^) that, after complete dissolution, is significantly below the concentration causing cytotoxicity (10 µg mL^−1^).

## 1. Introduction

Healthcare-associated infections are a significant concern in hospitals, contributing to increased mortality and morbidity. Of particular note is the attachment of the *Staphylococcus* spp. bacterial genus to a diverse array of textile materials, a genus ubiquitous on human skin [1]. Among the pathogens recognized by the World Health Organization responsible for healthcare-associated infections is *Staphylococcus aureus* (*S. aureus*) [2]. Microorganism colonization on textile surfaces has detrimental effects, including health problems, disease transfer, cross-infections, unpleasant odors, and reduced textile quality due to diminished fiber strength [3,4]. Medical textiles made from natural and synthetic fibers are used in healthcare for emergency treatment, as well as hygienic, surgical, and clinical purposes [5]. Textiles used for wound care, like adsorbent pads, wound contact layers, bandages, plasters, gauze, lint, and wadding are made of different fiber types ranging from natural materials, like cotton, to synthetic ones, like polyester [6]. Due to inherent hydrophilicity, natural materials like cotton are more susceptible to microbial colonization than hydrophobic synthetic materials such as polyester [7]. The development of coatings on fabric surfaces is a viable strategy to modulate hydrophobicity, paving the way for creating textiles with water-repellent properties [8]. Beyond altering hydrophobicity, utilizing organic molecules in coatings holds promise for deterring bacterial adhesion due to the repulsive force between negatively charged bacterial cells and coatings with a negative charge [9,10]. 

Nanomaterials have found extensive application in the production of functional textiles with diverse properties, such as UV protection, fire resistance, electrical conductivity, wrinkle resistance, self-cleaning, and antibacterial efficiency, all without compromising fundamental textile characteristics [11]. The development of antibacterial textiles is especially applicable in healthcare settings, including hospitals, nursing homes, and other environments susceptible to microbial infections or contaminations [5]. Metallic nanoparticles (NPs), including silver [12], copper [13], gold [14], graphitic carbon nitride [15], cerium oxide [16], iron oxide [16], titanium oxide [17], and zinc oxide [18] (among others [11]) have been employed for textile functionalization. Zinc oxide (ZnO) NPs, in particular, are extensively studied for their antibacterial potential and skin-friendly properties [19]. They have been integrated into various applications, such as air filters with antibacterial properties [20], antibacterial cotton materials for healthcare applications [21,22], textiles with photocatalytic self-cleaning and UV protection [23], and wound dressings [24]. ZnO NPs, synthesized through various methods, exhibit multiple activities, including antibacterial, antiviral, antifungal, antioxidant, anticancer, anti-inflammatory, wound healing, antidiabetic, and cardioprotective effects [25]. Furthermore, ZnO NPs have demonstrated substantial antibacterial activity against *S. aureus* and other bacteria [25,26].

Various organic molecules, including chitosan, alginate, starch, cyclodextrins, and cellulose, have been employed for nanoparticle stabilization and adhesion onto fabrics during textile functionalization [11]. Poly(allylamine hydrochloride), PAH, has been utilized in previous textile treatments, including PAH/graphene oxide [27] and PAH/poly(styrene sulfonate) [28] microcapsules for controlled release of bioactive molecules, or as a precursor for nanoparticle (NP) adhesion [21,29]. Cotton pretreated with long-chain polyamines has been used for textile treatment with ZnO NPs through direct mineralization and NP deposition [21,29]. Such treatment requires the preparation of a fresh reaction mixture for each new ZnO textile treatment. However, our approach advocates treating fabrics with PAH multilayers and pre-prepared ZnO NPs, offering the advantage of multiple uses of the same solutions for fabric treatment.

Additionally, ensuring biocompatibility with human tissue is crucial, especially in healthcare facilities, where direct contact with the human body is frequent. The current state-of-the-art treatment highlights the need to address ZnO NP biosafety by modifying NPs surface reactivity to enhance biocompatibility and minimize adverse effects [26]. Notably, the cytotoxicity of NPs tends to surpass that of dissolved ions released from NPs [30]. Previous studies using textile materials coated solely with copper oxide NPs demonstrated good antibacterial activity but exhibited high cytotoxicity toward human cells due to loosely bound NPs [31]. To mitigate such NP effects, using polyelectrolyte multilayers could limit NP release, allowing only the release of ions from the textile surface, thus retaining their antibacterial efficacy [32,33]. 

This study aimed to prepare novel nanohybrid PAH and ZnO multilayers on textile materials (cotton, nylon, and polyester) under ultrasonic treatment. The purpose was to create an active antibacterial coating against *S. aureus* by inactivating both planktonic and adhered *S. aureus* bacteria while preserving biocompatibility with human keratinocyte cells. Furthermore, this study aimed to identify the differences in antibacterial activity between coated cotton, nylon, and polyester. The analysis of the prepared ZnO NPs and the determination of coated textile surface characteristics through profilometry, contact angle, and electrokinetic analyzer measurements were conducted. Antibacterial experiments were performed using the *S. aureus* strain and biocompatibility with immortalized human keratinocyte cells. In conclusion, the findings of this research demonstrate that the novel nanohybrid PAH/ZnO coatings alter the hydrophobicity of textiles and exhibit antibacterial activity and biocompatibility. Among the used textiles, cotton demonstrated the highest antibacterial activity due to the higher amount of immobilized ZnO NPs. The PAH multilayers can be used for ZnO NP immobilization in amounts that cause antibacterial activity against *S. aureus* but still do not cause cytotoxicity against skin cells (keratinocytes). 

## 2. Materials and Methods

### 2.1. Synthesis of ZnO Nanoparticles

Rod-like ZnO nanoparticles (ZnO NPs) were synthesized using the analytical grade chemicals zinc acetate dihydrate (Zn(CH_3_CO_2_)_2_∙2H_2_O) and sodium hydroxide (NaOH), both of which were obtained from Sigma–Aldrich (St. Louis, MO, USA). The ZnO NPs were prepared by a modified preparation procedure [34]; namely, the Zn(CH_3_CO_2_)_2_ solution (*c*(Zn(CH_3_CO_2_)_2_) = 0.1 mol L^−1^) was prepared in a water solution, while ZnO NPs were obtained by adding NaOH (*c*(NaOH) = 5 mol L^−1^, 50 mL) in 150 mL of Zn(CH_3_CO_2_)_2_ solution. The resulting suspension was heated to 85 °C for two hours under magnetic mixing (300 rpm). The precipitate was centrifugated (6000 rpm), washed five times with deionized water, dried at 100 °C, and subsequently annealed at 400 °C for two hours (Nabertherm furnace, Bremen, Germany). Stabilization of ZnO NPs involved the addition of sodium poly(4-styrene sulfonate, PSS) (0.133 mL, *w* = 0.5%, Sigma–Aldrich, (St. Louis, MO, USA) to a 10 mL NP suspension (*γ*(ZnO NP_stock_) = 1500 µg mL^−1^). 

### 2.2. Fabrication of the Textile Coating

Nylon, cotton, and polyester samples were acquired from Pamigo Zagreb (Zagreb, Croatia). Textile samples (dimension 1 cm × 1 cm) underwent washing in an alkaline solution (*c*(NaOH) = 1 mol L^−1^) for 30 min, followed by rinsing with water. Subsequently, the samples were immersed for 15 min in a 3.0 mg mL^−1^ solution of poly(allylamine hydrochloride) (PAH, Sigma–Aldrich, St. Louis, MO, USA at pH = 7.5. The samples were then immersed in a ZnO NP suspension (115 µg mL^−1^) and sonicated for 5 min, followed by 10 min of soaking without sonication. This process was repeated until the surface was coated with three layers of ZnO NPs (Figure 1). The coating was finalized with a layer of PAH. After each PAH and ZnO NP layer, the textiles were rinsed with water and dried with warm air. 

### 2.3. Nanoparticle Characterization 

The composition of prepared ZnO NPs was determined using a Malvern Panalytical Empyrean X-ray diffractometer (Almelo, The Netherlands) in Bragg–Brentano geometry in the 2*θ* range 20°–80°, with a Cu-target tube and step size of 0.0131°, 1 s per step. The diffraction patterns were analyzed using PANalytical High Score Plus 3.0 software. Fourier-transform infrared (FTIR) spectroscopy, in the (340–4000) cm^−1^ range, was conducted on a PerkinElmer FT-IR C89391 (PerkinElmer, Waltham, MA, USA). NP morphology was determined using a Schottky field emission scanning electron microscope (SEM) Jeol JSM-7600F (Jeol Ltd., Tokyo, Japan). NP size distribution was determined from SEM images using ImageJ 2 software. The surface charge density of ZnO nanoparticles (ZnO NPs) was quantified using a Mütek-PCD05 instrument (BTG Instruments, Eclépens, Switzerland). A nanoparticle suspension [*γ*(ZnO) = 100 μg mL^−1^] was prepared in 70 mL of aqueous NaCl [*c*(NaCl) = 0.001 mol L^−1^] solution. The suspension underwent 1 min sonication to ensure homogeneity. The suspension pH was adjusted with NaOH [*c*(NaOH) = 0.1 mol L^−1^]. Suspensions were prepared at pH 7.5 and 10, as lower pH led to NP dissolution. A polyelectrolyte titrant solution of poly(diallyl dimethylammonium chloride) (*c*(PDDA) = 10^−4^ mol L^−1^) was added to the Mütek-PCD05 sample cell containing the NP suspension. Streaming potential signals were monitored during manual PDDA titration. The streaming potential was measured until the signal shifted from negative to positive when the neutralization of the NPs was achieved. The total charge *σ*_tot_ (C m^−2^) of NPs was computed by:
(1)$$\sigma_{tot}=\;\frac{F\;\times \;c\left(PDDA\right)\;\times \;V(PDDA)}{m\left(NPs\right)\;\times \;s}$$
where *F* (C mol^−1^) is the Faraday’s constant, *c*(PDDA) is the titrant concentration, *V*(PDDA) is the titrant volume needed to reach the point of zero charge, *m*(NPs) is the mass of NPs added to the suspension, and *s* (m^2^ g^−1^) is the specific surface area. The specific surface area was determined by nitrogen adsorption using an ASAP 2020 Sorber (Micromeritics, Norcross, GA, USA) at −196 °C, following the Brunauer–Emmett–Teller (BET) theory. Prior to testing, samples were outgassed under vacuum for 15 h at 120 °C. The sample mass in the analyzer was ≈0.2 g.

The photocatalytic activity of ZnO NPs was conducted as already published with small modifications [35]. Stock solutions of methylene blue (MB) 35 mg L^−1^ (Sigma–Aldrich, St. Louis, MO, USA), methyl orange (MO) 20 mg L^−1^ (Merck, Darmstadt, Germany), and basic fuchsine 18 mg L^−1^ (Sigma–Aldrich, St. Louis, MO, USA) were each prepared in 100 mL of water. Before the start of the experiments, equal volumes of each dye were mixed and 20 mg of ZnO NPs was added to 10 mL of mixed solution. The prepared suspension was magnetically stirred in dark conditions for 30 min to establish the absorption–desorption equilibrium of dyes on the surface of ZnO NPs. The suspension was irradiated in a closed chamber containing six UVA lamps corresponding to four 15W-UVA lamps (Philips Cleo; broad maximum at 355 nm) [36]. The aliquots of 2.0 mL were taken periodically at 0, 10, 60, 120, and 150 min, then centrifuged for 5 min at 6000 rpm. The supernatant was analyzed on a Varian Cary 50 Bio UV–Vis spectrophotometer (Agilent, Santa Clara, CA, USA) in a range of 300–700 nm.

### 2.4. Characterisation of the Textile Coating 

The hydrophobicity of uncoated textiles, textiles coated with PAH, and PAH/ZnO layers was determined using an Attension Theta tensiometer (Biolin Scientific AB, Gothenburg, Sweden). With the sessile-drop technique, the water contact angle was measured with a needle of 0.4 mm diameter used for water droplet seeding (5 µL) on the textile surfaces. The surface charge of the textiles was determined by a SurPASS electrokinetic analyzer (Anton Paar GmbH, Graz, Austria) at pH = 6 in 1 mmol L^−1^ KCl solution. The surface profile and roughness of the textile surfaces were analyzed by non-contact optical profilometry on the Zygo Zegage Pro optical profilometer HR (Zygo Corporation, Middlefield, CT, USA). The textiles were coated with 20 nm carbon and (5–10) nm platinum for the measurement. The morphology of the textiles was additionally analyzed with scanning electron microscopy (SEM) Jeol JSM-7600F (Jeol Ltd., Tokyo, Japan). The amount of released Zn^2+^ ions from the PAH/ZnO layers was determined using inductively coupled plasma mass spectrometry (ICP-MS). An Agilent 7900x Series ICP-MS instrument (Agilent Technologies, Tokyo, Japan) was equipped with an autosampler (SPS-4, Agilent Technologies, Tokyo, Japan), a glass Micromist nebulizer and a double-spray, Scott-type spray chamber made of quartz. Before measurement, triplicates of textile materials were immersed in 5 mL of deionized water and incubated for 24 h at room temperature. The solution above the samples was collected to determine the released Zn^2+^ ions. Also, the ICP-MS method in “single particle” mode was used to determine ZnO NPs released from the textile surfaces. Additionally, for the total amount of ZnO in the PAH/ZnO layers, triplicates of textile materials were acidified with hydrochloric acid [*c*(HCl) = 1 mol L^−1^] in 5 mL of deionized water until pH = 2 for total ZnO NP dissolution. The total ZnO NP mass on the 1 cm × 1 cm coated textile samples was calculated from the Zn^2+^ ions measured in the acidified solutions. 

### 2.5. Antibacterial Studies

The bacterial solution was prepared by transferring one *S. aureus* ATCC 25,923 colony into 5 mL of the brain heart infusion growth medium (BHI, Biolife Italiana Srl, Milan, Italy). The prepared suspension was incubated overnight at 37 °C. The optical density of the overnight bacterial culture was measured at 620 nm on a Tecan F200 Pro spectrophotometer (Tecan, Mannedorf/Zürich, Switzerland). The culture was diluted to 0.10 optical density at 620 nm in 5 mL of fresh BHI. The diluted suspension was incubated at 37 °C for 2 h to achieve exponential cell growth. The *S. aureus* suspension was then diluted in 0.9% NaCl to ≈10^5^ CFU mL^−1^. The 1 cm × 1 cm textile samples were sterilized under UV light and placed in a 24-well plate. Antibacterial testing on textile samples was conducted based on the ISO 20743 norm [37]. Additionally, 100 μL of the diluted bacteria suspension was loaded onto the samples in triplicate and incubated for 24 h at room temperature in dark conditions. Subsequently, the textile samples were washed two times with 1 mL of 0.9% NaCl solution, and the washing solution was collected. Washed textile samples were placed into 2.5 mL of 0.9% NaCl solution, sonicated in a sonication water bath for 5 min, and vigorously vortexed for 15 s at the highest speed. After a series of dilutions, 10 μL of the *S. aureus* suspension was loaded on plate count agar plates (incubated at 37 °C, 18 h) to measure the planktonic cells in the washing solution and adherent cells from the textile surface. 

The logarithmic reduction (L) after the antibacterial tests was calculated from Equation (2):
(2)$$L=\log_{10}(\frac{A}{B})$$
where A is the number of viable bacteria before coating and B is the number of viable bacteria after the coating application. From the logarithmic reduction, the percent reduction (P) was calculated from Equation (3):
(3)$$P\;=\;(1-\;\frac{B}{A})\times 100$$

The percentage reduction was used to determine the coating antibacterials efficiency. 

### 2.6. Biocompatibility Tests

Biocompatibility experiments were conducted on immortalized human keratinocytes HaCaT cells (ATCC, Manassas, VA, USA) following ISO 10993-5 standard [38]. Cells were cultured in 10 mL high glucose Dulbecco’s Modified Eagle Medium (DMEM) supplemented with 10% heat-inactivated fetal bovine serum (HI-FBS), 1% Non-Essential-Amino-Acids, and 1% Penicillin/Streptomycin (Gibco, Life Technologies Corporation, Paisley, UK) at 37 °C and 5% CO_2_ until 80% confluence. After removing the culture medium, HaCaT cells were rinsed with sterile phosphate-buffered saline (PBS). To detach the cells, 1 mL of 0.25% Trypsin-EDTA solution (Sigma–Aldrich, St. Louis, MO, USA) was added to the flask and incubated for 10 min at 37 °C and 5% CO_2_. The TC20 automated cell counter (Biorad, Hercules, CA, USA) was used to count detached cells. Subsequently, the cells were seeded in sterile 96-well plates (Thermo Fisher Scientific, Waltham, MA, USA) with a seeding density of 20,000 cells/well. The seeded 96-well plate was incubated for 24 h at 37 °C and 5% CO_2_. The cells adhered to the 96-well plates and were washed with PBS, covered with 100 µL of the media with Zn^2+^ ions (0 < *γ*(Zn^2+^) < 20) µg mL^−1^, and incubated for 4 h. A solution without Zn^2+^ ions was used as the negative control, while cells treated with 1% Triton-X were the positive control. After incubation with the Zn^2+^ ions, the cells were washed two times with PBS and covered with 20 μL of CellTiter 96^®^ (Aqueous One Solution Cell Proliferation Assay, Promega, Madison, WI, USA) and 100 µL of media. The plates were incubated for three hours at 37 °C and 5% CO_2_. Afterward, the solution absorbance at 490 nm was measured using a Clariostar^®^ plate reader (BMG LABTECH GmbH, Ortenberg, Germany). All experiments were conducted in sextuplicate.

### 2.7. Statistical Analysis

All experiments were conducted in triplicate (if not stated otherwise) and the Tukey test (GraphPad Prism 8, La Jolla, CA, USA) was used to analyze the results statistically. A two-way analysis of variance (ANOVA) was applied to evaluate the statistical significance of the textile type and PAH/ZnO coating antibacterial effect. Furthermore, the statistical significance of biocompatibility experiments was conducted with a one-way analysis of variance (ANOVA). The significance of the results was indicated according to *p*-values: * *p* ≤ 0.05, ** *p* ≤ 0.01, *** *p* ≤ 0.001, and **** *p* ≤ 0.0001. Statistical significance was considered for *p*-values below 0.05, i.e., 95% of the confidence interval.

## 3. Results and Discussion

### 3.1. Nanoparticle Characterization

The FTIR analysis (Figure 1a) showed specific vibration peaks in the region below 400 cm^−1^, which can be assigned to a Zn–O symmetric stretching vibration [39]. Furthermore, minor broad peaks in the regions ≈ 3400 cm^−1^ correspond to water and OH^−^ ions on the surface of ZnO NPs, while ≈1500 cm^−1^ correspond to C=O and carboxylate [40], and ≈870 cm^−1^ could correspond to sodium oxide, sodium hydroxides, and sodium carbonates remaining after the annealing of the sample in which, after the synthesis, a small amount of NaOH remained [41].

Distinct diffraction peaks corresponding to ZnO were identified in the XRD analysis (Figure 1b) at 2*θ* angles of 31.83°, 34.49°, 36.32°, 47.62°, 56.66°, 62.93°, 66.47°, 68.03°, 69.12°, 72.63°, and 77.00°, corresponding to ZnO crystallographic planes (*hkl*) (100), (002), (101), (102), (110), (103), (200), (112), (201), (004) and (202), respectively. Validation was performed using JCPDS card number 36-1451. The detected minor peaks were attributed to various sodium oxides, sodium hydroxides, and sodium carbonates remaining after the annealing of the ZnO sample at 400 °C, during which the remaining NaOH decomposed [41,42]. The amount of impurities was below 3% and could be disregarded as significant for the antibacterial application on textile fibers. 

Furthermore, the morphology of the ZnO NPs was determined with SEM analysis (Figure 1c). The ZnO NPs exhibited elongated rod-like morphology. Based on the SEM images, the ZnO NP size (Figure 1d) had an overall narrow distribution with an average size of (88.3 ± 27.0) nm (number of crystals 250). The ZnO NPs exhibited a surface area of 18.2 m^2^ g^−1^. Additionally, the prepared NPs exhibited a negative surface charge (Figure 1e) at both neutral and basic conditions, with a total surface charge (calculated from Equation (1)) of (0.33 ± 0.01) C m^−2^ at pH 7.5 and (0.49 ± 0.18) C m^−2^ at pH 10. Given the negative surface charge, the coating of textiles was ensured by applying positively charged PAH layers to facilitate the adhesion of nanoparticles to the textile surface. 

The photocatalytic activity of ZnO NPs was conducted on a mixture of three dyes, namely methylene blue, basic fuchsin, and methyl orange, showing absorption maximums at different wavelengths (Figure 1f). Following the addition of ZnO NPs to the dye mixture, photocatalytic activity was observed (Figure 1g). The adsorption maxima decreased with prolonged irradiation time. Similar results were obtained previously [35,43] demonstrating ZnO photocatalytic activity against dyes. ZnO NPs cause color fading due to the degradation of dyes in fabrics [44], which is unwanted in the textile industry. Due to the photocatalytic activity, the coating procedure with the produced ZnO NPs should be applied on fabrics intended for wound dressing and single-used fabrics where antibacterial properties are desirable and the exposure to light is not pronounced. 

### 3.2. Textile Surface Characterization

The coating procedure with PAH and ZnO NPs was conducted on various textile types, namely cotton, nylon, and polyester. First, the textiles were washed under alkaline conditions (NaOH) to remove impurities and potential grease traces. PAH polyelectrolyte is positively charged at neutral pH and can be electrostatically attracted to oxygen atoms on the textile surfaces [21]. After applying the first PAH layer, the textiles were incubated in ZnO NP solution under sonication. The ZnO NPs were applied between four PAH layers (Figure 1). The PAH terminating layer was used to prevent the release of the ZnO NPs from the textile surfaces. As shown in Section 3.1, the ZnO NPs exhibited a negatively charged surface at neutral conditions, which enables electrostatic adhesion to the positively charged PAH layer on the textile surface. To validate the development of the coating, changes in streaming potential (Figure 2a) and hydrophobicity (Figure 2b) of the textile surfaces were assessed. 

Streaming potential measurements (Figure 2a) on uncoated textiles, textiles coated with one PAH layer, and PAH/ZnO multilayers revealed a significant shift in surface zeta potential depending on the textile type (*p* < 0.0001) and coating procedure (*p* < 0.0001). The zeta potential (at pH = 6) of the uncoated textile was negative overall, and it was also overall lower for polyester (−224.9 ± 0.9) mV. After the coating with one layer of positively charged PAH, the zeta potential changed and was much higher than that on the uncoated fabric. The PAH-coated cotton and nylon became positive by charge, while the surface of the polyester was still slightly negative (−12.8 ± 1.8) mV. The significant change in the streaming potential for all textile types (*p* < 0.0001) confirms the adhesion of the positive PAH polyelectrolyte on the textile surfaces. Furthermore, after coating the textiles with multiple PAH/ZnO layers, the zeta potential became even more positive (*p* < 0.0001), being the highest for polyester (70.8 ± 4.4) mV and similar for cotton and nylon fabrics (≈25 mV). The overall surface charge impacts the adhesion of bacterial cells to material surfaces. *S. aureus* bacteria have a negative net surface charge, which is primarily attributed to teichoic acid [45] and phosphate heads in the cell’s lipid bilayer [46]. Repulsive electrostatic forces between negative bacterial cells and negatively charged surfaces can reduce bacterial adhesion [9,10]. Based on the measured textile surface charge (Figure 3, it would be expected that the highest bacterial adhesion would occur on more positively charged fabrics. 

The hydrophobicity of the textiles was determined by measuring the water contact angle on the uncoated textiles and textiles coated with PAH and PAH/ZnO multilayers (Figure 2b). The statistical analysis showed a significant difference in the hydrophobicity of the samples regarding the textile type (*p* < 0.0001) and coating procedure applied (*p* < 0.0001). The uncoated cotton and nylon fabrics had the lowest water contact angles and could be considered hydrophilic, while the polyester fabric showed hydrophobic properties. Two-way ANOVA was used to statistically determine the influence of the coating procedure. The results demonstrated a significant change in the hydrophobicity after the PAH coating of cotton (*p* = 0.0001) and nylon (*p* < 0.0001), while there was no statistical difference in the case of polyester (*p* = 0.9998). Similarly, after the PAH/ZnO coating, the hydrophobicity changed compared to uncoated textiles only in the case of cotton and nylon (*p* < 0.0001). After coating with one layer of PAH and PAH/ZnO multilayers, all fabrics exhibited slight hydrophobic properties with similar water contact angles (CA > 95°) regardless of the textile type used. Bacterial colonization on textile surfaces depends on the textile material’s hydrophobic and hygroscopic properties [47]. Numerous studies have shown that bacteria with hydrophilic surface properties adhere more strongly to hydrophilic surfaces [48,49,50,51]. The bacterial adhesion could be lowered by enhancing the hydrophobicity, especially in the case of cotton and nylon. 

Furthermore, the surface roughness and fiber arrangement could lead to different outcomes during the coating procedure due to the larger available area. The surface roughness of the textiles was determined by optical profilometry (Figure 2c), and the results showed that the surface roughness was the highest for nylon (90.6 ± 0.3) µm, while the lowest was for cotton (34.2 ± 0.1) µm fabric. Cotton had a smoother overall surface, with fewer deep trenches between the fibers. The nylon and polyester fibers had peak-to-valley numbers that were significantly higher, and the significant difference between polyester and nylon is seen in the bundle arrangement. The same surface characteristics can also be observed by SEM images (Figure 3). The fiber arrangement was different for each fabric. Cotton exhibited densely intertwined fibers with several loose threads on the surface and empty spaces between fiber bundles. Polyester fibers were neatly packed in interwoven fiber bundles. In the case of polyester fabric, the fiber bundles were more tightly woven without free space. On the other hand, nylon textiles exhibited an irregularly disordered arrangement of threads without defined fiber bundles due to loose threads. The polyester and nylon fibers also exhibited smoother fiber surfaces and thicker threads than cotton.

Furthermore, a difference in the amount of ZnO NPs on the textile can be observed in the SEM images (Figure 3, second row). A higher number of ZnO NPs were present on cotton than on polyester and nylon fibers. EDS mapping was performed to confirm the composition of the particles on the textile surfaces (Figure 3, third row). The results confirmed the presence of a higher amount of ZnO NPs on the surface with cotton fibers. 

An ICP-MS analysis was performed to confirm a difference in the amount of ZnO NPs on the coated textiles (Figure 4a). To determine the total ZnO amount on the coated textiles, the samples were incubated in an acidified solution (pH = 2) for 24 h. The results demonstrate a significantly higher amount of Zn^2+^ ions on the cotton samples (4.65 ± 0.14) µg mL^−1^ than on polyester (0.50 ± 0.02) µg mL^−1^ and nylon (0.32 ± 0.01) µg mL^−1^ after the complete dissolution of ZnO NPs. Based on the released Zn^2+^ ions, the amount of ZnO NPs on the cotton was (28.9 ± 0.87) µg cm^−2^, polyester (3.1 ± 0.09) µg cm^−2^, and nylon (2.0 ± 0.06) µg cm^−2^. SEM-EDS and ICP-MS analyses show that the cotton sample is a better candidate for treatment with PAH/ZnO layers for possible antibacterial application, as a higher amount of ZnO NPs could be embedded in the PAH multilayers. 

Furthermore, to determine the extent of Zn^2+^ ions released from the fabrics in ambient conditions, the PAH/ZnO-coated textiles were incubated in water for 24 h. The ICP-MS analysis (Figure 4b) demonstrated that the amount of Zn^2+^ ions released was the highest for cotton (0.99 ± 0.03) µg mL^−1^, followed by polyester (0.51 ± 0.02) µg mL^−1^, and nylon (0.31 ± 0.01) µg mL^−1^. Additionally, the solutions were screened for the presence of ZnO NPs using the ICP-MS method in “single particle” mode. No ZnO-containing nanoparticles were detected. The results show that the PAH layers prevent the release of ZnO NPs from the coated textiles. Such surfaces are more beneficial for human usage, as NPs cause a higher cytotoxic effect on human cells than dissolved ions [30]. 

### 3.3. Antibacterial Activity

Antibacterial tests were performed with the *S. aureus* ATCC 25,923 strain, a common bacterium associated with textile-related infections [1,4]. In the experiments, the number of planktonic and adhered *S. aureus* was assessed on both uncoated textile surfaces and those coated with PAH and PAH/ZnO multilayers (Figure 5).

Antibacterial experiments with planktonic *S. aureus* (Figure 5a) showed no statistically significant variance between fabric types (*p* = 0.0874). However, a significant difference was observed between uncoated and coated surfaces (*p* < 0.0001). A two-way analysis of variance (ANOVA) was employed to examine statistical differences between textile and coating types regarding *S. aureus* cell viability. The number of planktonic cells did not significantly differ between uncoated and PAH-coated textiles (*p* = 0.1018). Following the application of a PAH/ZnO coating, the number of viable planktonic cells decreased significantly, and, compared to uncoated textiles, the PAH/ZnO layers reduced the number of viable planktonic *S. aureus* on cotton (*p* < 0.0001), nylon (*p* = 0.0005), and polyester (*p* < 0.0001). The reduction in viable planktonic *S. aureus* cells on PAH/ZnO coated cotton, nylon, and polyester compared to uncoated textiles exceeded 99% (Table 1). 

Regarding the number of adhered cells (Figure 5b), significant variations were observed between fabric types (*p* < 0.0001) and uncoated/coated textile surfaces (*p* < 0.0001). Two-way ANOVA indicated a statistically significant decrease in the number of viable adhered cells following treatment with PAH in the case of cotton (*p* = 0.0433); however, no significant difference was observed for nylon and polyester (*p* > 0.8940). The literature suggests potential antibacterial properties in poly(allylamine hydrochloride) [52,53,54]. The variability in PAH antibacterial efficacy among textiles may stem from differing PAH quantities. Cotton fibers composed of cellulose [55] possess more free hydroxyl groups compared to nylon fibers, which are made from polymeric amides [56]. Polyester fibers composed mainly of esters [57] also exhibit different surface characteristics. PAH, which contains positively charged amine groups, electrostatically adheres to oxygen atoms on textile surfaces [21]. The higher presence of hydroxyl groups and oxygen atoms in cotton fabric likely leads to better coverage with PAH compared to nylon and polyester. Compared to uncoated cotton, the number of viable *S. aureus* cells on PAH-coated cotton was reduced by (90.12 ± 6.81)% (Table 1).

Upon PAH/ZnO NP application, the number of viable adhered cells decreased significantly on cotton (*p* < 0.0001), nylon (*p* = 0.0001), and polyester (*p* < 0.0001). Compared to uncoated textiles, the number of viable adhered *S. aureus* cells on PAH/ZnO coated cotton, nylon, and polyester was reduced by more than 99% (Table 1). Statistical analysis indicated that cotton fabric coated with PAH/ZnO layers exhibited a more pronounced antibacterial effect against *S. aureus* than nylon and polyester (*p* = 0.0004). The higher antibacterial efficiency of cotton samples may be attributed to the increased release of Zn^2+^ ions from the fabric surface (Figure 4b). Additionally, the lower surface charge of PAH/ZnO coated textiles may lead to reduced bacterial adhesion.

Despite the lower amount of ZnO NPs on coated nylon and polyester, a significant reduction in both planktonic and adhered *S. aureus* was observed (Figure 5, Table 1), suggesting that the attached ZnO NPs on nylon and polyester are sufficient for inducing a high antibacterial effect. PAH/ZnO-coated textiles exhibited antibacterial characteristics, resulting in a 2–3-log reduction (Table 1) in viable planktonic and adhered *S. aureus* cells, surpassing the predetermined threshold for materials to be considered antibacterial [58]. Overall, when comparing all three textile types, cotton emerges as a more suitable material for PAH/ZnO coating development due to better adherence of ZnO NPs and slightly higher antibacterial performance.

### 3.4. Biocompatibility Tests

The in vitro biocompatibility was assessed using immortalized human keratinocyte (HaCaT) cells. Potential toxicity from Zn^2+^ ions was evaluated through an indirect assay measuring cell viability. The change in metabolic activity was observed within a concentration range from 0.5 µg mL^−1^ to 20 µg mL^−1^ of Zn^2+^ ions. The metabolic activity results (Figure 6) revealed no statistically significant decrease in cell viability up to 7 µg mL^−1^ of Zn^2+^ ions compared to the control specimen without Zn^2+^ ions (*p* > 0.05). ICP-MS data (Figure 4b) indicated the highest release of Zn^2+^ ions into the water for cotton (0.99 ± 0.03) µg mL^−1^, with a total potential release after complete ZnO dissolution being (4.65 ± 0.14) µg mL^−1^.

Biocompatibility assessment demonstrated a decrease in viability only at the highest tested Zn^2+^ ion concentration (Figure 6), namely 10 µg mL^−1^ (*p* = 0.0054) and 20 µg mL^−1^ (*p* < 0.0001). Notably, the amount of Zn^2+^ ions on the coated textiles remained significantly lower than the threshold concentration (10 µg mL^−1^) associated with HaCaT cell cytotoxicity. This finding supports the conclusion that the coated textiles are biocompatible, as the Zn^2+^ ion levels (*γ*(Zn^2+^) = (4.65 ± 0.14) µg mL^−1^) are well below those inducing cytotoxic effects. Contrastingly, previous studies on orthotic materials treated with copper oxide NPs demonstrated antibacterial efficacy against Gram-positive bacteria in addition to induced cytotoxic effects on human cells due to loosely bound NPs [31]. To mitigate the NPs’ cytotoxicity, the current study employed PAH polyelectrolyte multilayers to immobilize the NPs, allowing for controlled release of Zn^2+^ ions. 

Overall, the results demonstrate that, in the case of both planktonic and adhered bacterial cells, there was no statistical difference in the number of viable cells between uncoated cotton, nylon, and polyester fabrics regardless of their surface characteristics. Even though the fabrics demonstrated significantly different zeta potentials, hydrophobicity, and surface roughness (Figure 2) during 24 h of incubation, the bacterial growth was not influenced. This demonstrates that the adhesion of bacteria, which is often lowered due to electrostatic repulsion between negative surfaces and negatively charged bacteria [9,10] as well as the slower adhesion of hydrophilic bacteria to hydrophobic surfaces [48,49,50,51], was not affected by the fabric surface properties. With prolonged incubation (24 h), the bacterial solution gets in between textile fibers, and the influence of the surface properties on the bacterial adhesion and growth is not expressed. However, the composition of fibers does influence the development of PAH/ZnO multilayers, which lower the viability of bacterial cells by more than 99% (Table 1). The higher presence of oxygen atoms and hydroxyl groups in cotton fabric [55] enables better electrostatic adherence of positively charged amine groups [21], resulting in superior adherence of negatively charged ZnO NPs (Figure 1e). As a result, the amount of ZnO NPs was significantly higher on the cotton sample (Figure 4a) and caused higher antibacterial efficiency (Figure 5b). This was expected, as it was previously shown that a higher amount of ZnO NPs deposited on cellulose materials caused more expressed antibacterial activity [59]. Furthermore, the biocompatibility (Figure 6) experiments demonstrate that the amount of Zn^2+^ on the treated textiles (Figure 4a) is well below the cytotoxic concentration (10 µg mL^−1^). 

The ICP-MS analysis demonstrated that ZnO NPs do not get detached from the PAH multilayers into the water solution during 24 h incubation. Since the NPs are immobilized on the textile surfaces and the experiments were conducted in the dark, it can be presumed that the antibacterial activity will mostly be expressed due to the released Zn^2+^ ions, slight H_2_O_2_ production, and surface oxygen vacancies on the ZnO NPs [60,61]. As previously demonstrated, the production of small amounts of H_2_O_2_ from the ZnO surface can occur in dark conditions by converting H_2_O into H_2_O_2_ by surface oxygen vacancies [60]. Along with Zn^2+^ ions released from NPs, the released H_2_O_2_ can cause significant antibacterial effects [59]. Zn^2+^ ions cause bacterial cell death through protein mis-metallation and dysfunction and reactive oxygen species generation, which cause lipid peroxidation, nucleic acid damage, and protein oxidation [32,62]. Zn^2+^ ions cause higher membrane permeability, resulting in higher cell susceptibility to the action of NPs [63]. In planktonic cells, direct interaction with ZnO NPs can be excluded as the ICP-MS measurements demonstrated that NPs do not get released from the PAH multilayers to the solution. On the other hand, bacteria attached to the surface could come into contact with NPs, leading to additional alternation in the cell energy metabolism by increasing sugar metabolism and pyrimidine biosynthesis and decreasing amino acid synthesis, as shown by Kadiyala et al. [32]. Employing immobilization techniques (such as immobilizing NPs in PAH layers) can enhance the safety profile of textiles while retaining their antibacterial properties. The toxicity of NPs is much higher than that of the dissolved ions [30], and careful control of exposure route, size [64], and surface modifications is crucial for maximizing antibacterial and minimizing NP toxic effects [65]. 

The development of antibacterial and biocompatible textiles using the proposed nanohybrid PAH/ZnO coating holds significant promise for diverse applications, particularly in healthcare. The usage of NPs in antibacterial applications must be conducted to preserve biocompatibility and safety for the user. The proposed method of textile coating enables the immobilization of ZnO NPs in an amount (Figure 4) that causes significant antibacterial activity while also being significantly below the concentrations (*γ*_dissolved_(Zn^2+^) = 10 µg mL^−1^) causing cytotoxicity. The PAH/ZnO coated textiles could be employed in textiles used for wound dressing and single-time fabrics used in contact isolation and operating rooms to prevent cross-contamination with pathogens. This could contribute to reducing infections, a critical concern in healthcare settings.

## 4. Conclusions

A comprehensive investigation into the development of antibacterial, and biocompatible textiles coated with PAH (poly(allylamine hydrochloride)) and ZnO NPs has provided valuable insights into their physicochemical properties and functional performance for the first time. The coating process altered the textile surfaces, impacting their surface charge and hydrophobicity. Streaming potential measurements and water contact angle analyses confirmed the successful formation of PAH and PAH/ZnO multilayers, which is crucial for subsequent antibacterial assessments. When comparing natural cotton and synthetic polyester and nylon fibers, the cotton fibers had 10 times higher amounts of adsorbed ZnO NPs on the surface after the coating procedure. The proposed textile coating procedure enables, firstly, the immobilization of ZnO NPs in PAH multilayers, which prevents the release of NPs from the textile surface. Secondly, the proposed procedure enables remarkable antibacterial efficacy, with a 2–3-log reduction in viable cells against both planktonic and adhered *S. aureus* cells. The substantial reduction in *S. aureus* viability, at >99%, on coated cotton, nylon, and polyester, shows excellent potential for antibacterial application. Cotton, due to its better ZnO NP adherence, demonstrated a slightly higher antibacterial performance. Thirdly, the coating procedure enables the binding of ZnO NPs in an amount (<30 µg cm^−2^) that, after complete dissolution, is significantly below the concentration causing cytotoxicity. Cytotoxic experiments show that concentrations of released Zn^2+^ below 10 ug mL^−1^ can be considered safe and will not cause cytotoxicity against keratinocyte cells, the most common type of skin cell and structural component of the epidermis. Due to the ZnO NPs photocatalytic activity against dyes and the dissolution of ZnO to Zn^2+^ ions, the fabrics coated with the proposed coating method are suitable for textiles intended for wound care and single-used fabrics where antibacterial properties are desirable and the exposure to light is not pronounced.

Due to the increasing use of metal NPs for antibacterial purposes, their cytotoxicity during application must be tested to ensure safe use for the patients. Our future research will be focused on the development of coatings that enable multiple uses of treated textiles, with an emphasis on maintaining the amount of NPs that will be safe for the user/patients and that will not cause cytotoxicity and adverse effects.

## Data Availability

The data that support the findings of this study are available from the corresponding authors upon reasonable request.

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
