# Peer review of "Poly(Allylamine Hydrochloride) and ZnO Nanohybrid Coating for the Development of Hydrophobic, Antibacterial, and Biocompatible Textiles"

_nanomaterials, 2024, doi:10.3390/nano14070570_

Round 1

Reviewer 1 Report

Comments and Suggestions for Authors

Mlinaric et al. have aimed to develop a novel multilayer coating with poly(allylamine hydrochloride) and ZnO nanoparticles on different textile materials and evaluate their antibacterial activity against S. aureus and biocompatibility with immortalized human keratinocyte cells. This is an interesting piece of work carried out systematically and well-written. However, the following issues need to be addressed before this article can be accepted for publication in Nanomaterials.   

1.    Keywords – “PAH-poly(allylamine hydrochloride)” should be corrected as “poly(allylamine hydrochloride)”.

2.    A reference citation each for the procedures described in sections 2.1, 2.2, 2.5, and 2.6 should be included.

3.    The 2 Theta values should be followed by a degree symbol without any space in between.

4.    In the equations 1 and 2, the comma at the end should be deleted.

5.    In the equation 3, the dot symbol should be replaced with a multiplication symbol.

6.    Why the stability of the prepared nanohybrid was not determined?

7.    A new paragraph discussing how the characteristics data obtained for nanohybrid and textiles influence the antibacterial and biocompatibility results should be included before the conclusion. This paragraph should also discuss the mechanism by providing a schematic diagram.

Comments on the Quality of English Language

Minor editing of English language required

Author Response

Reviewer #1

Mlinaric et al. have aimed to develop a novel multilayer coating with poly(allylamine hydrochloride) and ZnO nanoparticles on different textile materials and evaluate their antibacterial activity against S. aureus and biocompatibility with immortalized human keratinocyte cells. This is an interesting piece of work carried out systematically and well-written. However, the following issues need to be addressed before this article can be accepted for publication in Nanomaterials. 

We thank the reviewer for all the comments.

Our answers are as follows:

Keywords – “PAH-poly(allylamine hydrochloride)” should be corrected as “poly(allylamine hydrochloride)”.

The correction from “PAH-poly(allylamine hydrochloride)” to “poly(allylamine hydrochloride)” was made in the Keywords in line 40.

A reference citation each for the procedures described in sections 2.1, 2.2, 2.5, and 2.6 should be included.

Reference for section 2.1 (line 123) was added which we modified to obtain a different ZnO morphology than the one published before. Furthermore, the references to ISO norms were added to sections 2.5 (line 227) and 2.6 (line 244). There is no reference for section 2.2. because to the best of our knowledge, the treatment of textiles with multilayers of ZnO and PAH presented in this manuscript was not done before.

The 2 Theta values should be followed by a degree symbol without any space in between.

The value was corrected in the manuscript (lines 152-154) and now the Theta value is followed by the degree symbol without any space.

In the equations 1 and 2, the comma at the end should be deleted.

The comma after equations 1 and 2 was deleted.

In the equation 3, the dot symbol should be replaced with a multiplication symbol.

The dot symbol was changed to the multiplication symbol.

Why the stability of the prepared nanohybrid was not determined?

The ICP-MS measurements demonstrated that in the case of nylon and polyester fabric, all of the ZnO NPs get dissolved in water during 24 h incubation. This can be seen from the difference in the Zn2+ concentrations after complete ZnO dissolution with acid (Figure 4a) and release in water (Figure 4b). In the case of cotton, there is still some amount of ZnO NPs left on the fabric after 24 h incubation in water. This shows that the fastness of ZnO will be overcome during washing and most likely get removed completely. However, even though ZnO NPs get dissolved to Zn2+ ions PAH multilayers inhibit the release of ZnO NPs which lowers the cytotoxicity of the ZnO NPs towards human cells but lowers the viability of bacterial cells by > 99 %. Due to the dissolution, these fabrics can be used for single-use applications or the coating should be applied after each washing (which is most probably not optimal, time-consuming, expensive, etc.). Due to this, we changed the application of the textiles to wound dressings and fabrics intended for single use throughout the whole manuscript (lines 50-54, 566-568).

A new paragraph discussing how the characteristics data obtained for nanohybrid and textiles influence the antibacterial and biocompatibility results should be included before the conclusion. This paragraph should also discuss the mechanism by providing a schematic diagram.

The following paragraphs were added or rearranged in lines 517-569:

“Overall, the results demonstrate that in the case of both planktonic and adhered bacterial cells, there was no statistical difference in the number of viable cells between uncoated cotton, nylon, and polyester fabrics regardless of their surface characteristics. Even though the fabrics demonstrated significantly different zeta potentials, hydrophobicity, and surface roughness (Figure 2) during 24 h of incubation the bacterial growth was not influenced. This demonstrates that the adhesion of bacteria, which is often lowered due to electrostatic repulsion between negative surfaces and negatively charged bacteria [9,10] and slower adhesion of hydrophilic bacteria to hydrophobic surfaces [45–48], was not affected by the fabric surface properties. With prolonged incubation (24 hours) the bacterial solution gets in between textile fibers and the influence of the surface properties on the bacterial adhesion and growth is not expressed. However, the composition of fibers does influence the development of PAH / ZnO multilayers which lower the viability of bacterial cells by more than 99 % (Table 1). The higher presence of oxygen atoms and hydroxyl groups in cotton fabric [52] enables better electrostatic adherence of positively charged amine groups [21] resulting in better adherence of negatively charged ZnO NPs (Figure 1d). As a result, the amount of ZnO NPs was significantly higher on the cotton sample (Figure 4a) and caused higher antibacterial efficiency (Figure 5b). This was expected since previously it was shown that a higher amount of ZnO NPs deposited on cellulose materials caused more expressed antibacterial activity [56]. Furthermore, the biocompatibility (Figure 6) experiments demonstrate that the amount of Zn2+ on the treated textiles (Figure 4a) is well below the cytotoxic concentration (10 µg mL−1).

The ICP-MS analysis demonstrated that ZnO NPs do not get detached from the PAH multilayers into the water solution during 24 h incubation. Since the NPs are immobilized on the textile surfaces and the experiments were conducted in the dark, it can be presumed that the antibacterial activity will mostly be expressed due to the released Zn2+ ions, slight H2O2 production, and surface oxygen vacancies on the ZnO NPs [57,58]. As previously demonstrated the production of small amounts of H2O2 from the ZnO surface can occur in dark conditions by converting H2O into H2O2 by surface oxygen vacancies [57]. Along with Zn2+ ions released from NPs, the released H2O2 can cause significant antibacterial effects [56]. Zn2+ ions cause bacterial cell death through protein mis-metallation and dysfunction and reactive oxygen species generation, which cause lipid peroxidation, nucleic acid damage, and protein oxidation [32,59]. Zn2+ ions cause higher membrane permeability, resulting in higher cell susceptibility to the action of NPs [60]. In planktonic cells, direct interaction with ZnO NPs can be excluded since the ICP-MS measurements demonstrated that NPs do not get released from the PAH multilayers to the solution. On the other hand, bacteria attached to the surface could come in contact with NPs. This could lead to additional alternation in the cell energy metabolism by increasing sugar metabolism and pyrimidine biosynthesis and decreasing amino acid synthesis, as shown by Kadiyala et al. [32]. Employing immobilization techniques (such as immobilizing NPs in PAH layers) can enhance the safety profile of textiles while retaining their antibacterial properties. The toxicity of NPs is much higher than that of the dissolved ions [30], and careful control of exposure route, size [61], and surface modifications is crucial for maximizing antibacterial and minimizing NP toxic effects [62].

The development of antibacterial, and biocompatible textiles using the proposed nanohybrid PAH/ZnO coating holds significant promise for diverse applications, particularly in healthcare. The usage of NPs in antibacterial applications must be conducted to preserve biocompatibility and safety for the user. The proposed method of textile coating enables the immobilization of ZnO NPs in an amount (Figure 4) that causes significant antibacterial activity, while also being significantly below the concentrations (γdissolved(Zn2+) = 10 µg mL−1) causing cytotoxicity. The PAH/ZnO coated textiles could be employed in textiles used for wound dressing and single-time fabrics used in contact isolation and operating rooms to prevent cross-contamination with pathogens. This could contribute to the reduction of infections, a critical concern in healthcare settings.”

However, since the conducted research did not specifically investigate which mechanism of Zn2+ ions influences the metabolism of the bacteria and causes cell death we did not include a schematic diagram since it could be misleading and our study did not focus on the mechanism of Zn2+ action. The aim of the study was the development of a coating for immobilization of ZnO NPs on the surface of textiles, thus lowering the cytotoxicity toward human cells, but preserving excellent antibacterial activity.

Comments on the Quality of English Language

Minor editing of English language required

The manuscript was thoroughly checked, and the errors were corrected.

Reviewer 2 Report

Comments and Suggestions for Authors

The article "Poly(allylamine hydrochloride) and ZnO Nanohybrid Coating for the Development of Hydrophobic, Antibacterial, and Biocompatible Textiles" describes the obtaining of cotton, nylon and polyester fabrics with deposited ZnO nanoparticles and antibacterial activity of the obtained coated textiles. It is a valuable study that can be published after authors address the following problems:

Abstract should be checked and revised carefully by briefly introducing the work plan and key findings. Abstracts should highlight the innovation of the article, as often abstract section is presented separately in search engines, it must be able to stand alone as an informative piece. The key quantitative data showing the antibacterial activity should be included in the abstract.

This work is interesting and can be boosted further. Hence the following literature could prove this manuscript doi: 10.1016/j.arabjc.2018.12.003; doi: 10.3390/pharmaceutics14122842 and help depict the possible killing mechanisms. Additionally, please asses the use of antimicrobial and antibacterial terms across the manuscript (authors have tested only antibacterial properties).

The fastness of ZnO coating should be given, such as water washing resistance, friction resistance, etc. If the authors intend to use such materials for wound dressing for example, maybe the washing is no longer required. But for hospital gowns, bed linens etc, the resistance of coating to washing is important.

One fundamental issue that authors should address is the strong photocatalytic activity of ZnO vs organics. As fabric are usually dyed, or even the white ones in this case are covered with poly(allylamine hydrochloride), and the textiles are coming in contact with light, the presence of ZnO NPs can degrade the organics (dyes and polyamines) very quickly (reference to doi: 10.1016/j.ceramint.2022.11.178; doi: 10.3390/ijms24065677). As such the authors should propose a way to decrease photocatalytic activity (decrease ROS production) of ZnO NPs, even if this might decrease the antimicrobial activity, as ROS production is one of the several mechanisms involved.

As a secondary solution such treated textile could be used as white fabrics in underwear, wound dressing etc. (where is less light), and in single use applications (like wound dressing).

For every day cloths, that are usually coloured, how do authors propose to coordinate the sizing and dyeing process of fabric with the coating of this research?

Please recheck the precursor used for ZnO NPs. Diethylzinc “Zn(CH3CH2)2∙2H2O” is not crystalline at room temperature to come with two water molecules (is a pyrophoric liquid, that reacts with water, and comes either as pure liquid or as a solution – Sigma lists solutions in hexane, toluene or heptane). As it reacts with water, did authors really made a solution with over 80% water (see https://doi.org/10.1016/0022-328X(84)85004-4)? Or perhaps is just a typo and authors used zinc acetate dihydrate Zn(CH3CO2)2∙2H2O? Therefore it is advisable to present the names of chemicals.

The usual range for FTIR spectra is 400-4000 cm-1 as dropping under 400 cm-1 require expensive CsI optics (row 151). How did authors gather data from 340 cm-1 when spectrum was measured from 400 cm‑1? The ref 33 (doi: 10.1039/d1nj00727k) that authors use to explain the ZnO FTIR spectra presents peaks from 464 cm-1 upwards, for ZnWO4. Please explain and use more suitable references like those indicated above, with ZnO.

At XRD section, row 265, the 2theta values must be rechecked. 34.30o (101) value is wrong, as is listed after 34.40o (002) – should be 36-37o. The value for (201) is wrong (authors declare 70.00o at row 266) when from figure 1b it is clear that the value is smaller (should be around 69o). Please measure them exactly. Additionally, all Miller indices should be given in round brackets like above.

Minor peaks are visible in XRD, and the FTIR spectrum also has additional peaks at ~ 800, 1400, 3400 cm-1. These indicates existence of some impurities in ZnO – please clarify.

In figure 1a (ZnO FTIR) please use standard ticks of horizontal axe e.g. 500/1000/1500/2000 etc.

The conclusion should reflect the heuristic of the study. How is this system a better one? Conclusion section must be reworked to underline the novelty and advantages of this research, with actual numbers. Please highlight the salient findings and future perspective.

Author Response

Reviewer #2

Round 2

Reviewer 2 Report

Comments and Suggestions for Authors

The authors have responded to my comments and have addressed all my concerns, substantially improving the manuscript, therefore, I suggest publishing the paper titled "Optimizing Antimicrobial Efficacy: Investigating the Impact of Zinc Oxide Nanoparticle Shape and Size" in the current form.

Some minor corrections can be made during proofing phase (row 123 Zn(CH3OH2)2 should be Zn(CH3CO2)2 )